# Current Issues and Developments in Cyanobacteria-Derived Biofuel as a Potential Source of Energy for Sustainable Future

Kshetrimayum Birla Singh [1], Kaushalendra [2], Savita Verma [3], Rowland Lalnunpuii [4] and Jay Prakash Rajan [5,*]

1. Department of Zoology, School of Life Sciences, Manipur University, Canchipur 795003, India; birla.kshetri@gmail.com
2. Department of Zoology, Pachhunga University College, Aizawl 796001, India; kaushalpuc@gmail.com
3. Applied Sciences Department, Galgotias College of Engineering and Technology, Greater Noida 201306, India; savitanit@gmail.com
4. Department of Biotechnology, Pachhunga University College, Aizawl 796001, India; rowlandlalnunpuii@gmail.com
5. Department of Chemistry, Pachhunga University College, Aizawl 796001, India
* Correspondence: jaypr33@gmail.com

**Abstract:** Biofuel production using cyanobacteria aims to maintain the sustainability of an ecosystem with minimum impact on the environment, unlike fossil fuels, which cause havoc on the environment. The application of biofuel as an alternative energy source will not only help in maintaining a clean environment and improving air quality but also decrease harmful organic matter content from aquatic bodies. Cyanobacteria are valuable sources of many novel bioactive compounds, such as lipids and natural dyes, with potential commercial implications. One of the advantages of cyanobacteria is that their biochemical constituents can be modified by altering the source of nutrients and growth conditions. Careful changes in growth media and environmental conditions altering the quality and quantity of the biochemicals and yield capacity have been discussed and analyzed. In the present review, the challenges and successes achieved to date in the commercial production of biofuel and its application in the transportation industry are discussed. The authors also focus on different types of feedstocks obtained from biomass, especially from cyanobacterial species. This review also discusses the selection of appropriate cyanobacterial species with merits and demerits in the post-harvesting process. In sum, the current review provides insight into the use of organic bioresources to maintain a sustainable environment.

**Keywords:** algal biomass; algal oil feedstocks; phytoremediation; bioresource; clean energy; fatty acids

## 1. Introduction

In technical terms, fossil fuels include coal, oil, natural gas, and hydrocarbons. Humans have been using fossil fuels since time immemorial. However, since the debut of the Industrial Revolution in the 18th century, the use of fossil fuels, especially coal, has expanded beyond the imagination. After the formation of fossil fuels deep inside the crust millions of years ago, its use was sped up only a few hundred years back when one person named James Watt invented the steam engine. The extensive use of fossil fuels has altered the socio-economic status and has a negative impact on our natural environment. In the past few years, the reduction in the availability of fossil fuels has ignited the global scientific community to move toward renewable energy sources, which are sustainable and eco-friendly. Renewable energy sources include wind power, hydropower, hydroelectricity, nuclear energy, solar energy, geothermal energy, and biofuel. In the present review, the relevance and future prospect of biofuel production are addressed thoroughly. Biomass and microorganisms have the capacity to convert starting material from $H_2O$ and $CO_2$ in

the presence of solar energy to end-product intermediate metabolic compounds, which are directly used in biofuel products.

$$CO_2 + \text{solar energy} \rightarrow H_2O + \text{product} + O_2$$

During biofuel production, biomass is transformed into ethanol and biodiesel in a limited time span with the help of biological/nonbiological agents. One of the best biological agents is "microbial agents," which have drawn the attention of researchers worldwide [1–3]. Although humans are in the very early stage of the Type I civilization, our ever-increasing population and its dependency on machines have created significant demands for energy to run our modern and fast-growing global economies in the present and future [4–7]. The demerits of fossil fuels are their limited presence beneath the Earth's crust, the production of GHGs in the environment [8–11], and the increasing cost of refining procedures, which in turn create fluctuations in the global price of petroleum products. Under such a scenario, a new generation of biofuel may play a crucial role in decreasing our dependency on conventional fossil fuels as well as curtail the global production of greenhouse gases (GHGs) in the environment. Hence, researchers around the globe are investing their efforts into energy fuels being sustainable, renewable, and eco-friendly [12–20]. Among the various microbiota, cyanobacteria have the potential to become a promising device for the production of the next generation of biofuel. They have an outstanding reputation in the biosphere because of their key ability to fix nitrogen and carbon [21–23]. Only recently, scientists have developed advanced techniques to isolate and identify the valuable secondary metabolites that may be used in biotechnological applications to solve environmental issues. Cyanobacterial metabolism can be easily applied to technical innovations and, thus, is practically the most suitable candidate for the large-scale commercial production of biofuel [24–28].

The best part of cyanobacteria-driven biofuel production is the zero release of pollutants, whereas harmful emissions are one of the most severe drawbacks of conventional energy sources. Because of these unmatchable properties, cyanobacteria have emerged as the best microbial candidate for biofuel production [29–31]. Moreover, several cyano-metabolites are precursors for producing various hydrocarbons in biofuel. In addition, cyano-metabolites are used in wastewater treatment and microalgal biorefinery, collectively known as phytoremediation. In the present review, the major effect of nutrient stress derived from wastewater and its utilization in altering the metabolite contents in algal cells is discussed, stressing the cost-benefit ratios in algal industries. Furthermore, an overview of The latest research works to initiate a biochemical way to extract and transesterify cyanobacteria to produce valuable biofuel is included. Finally, this review discusses the benefits of cyanobacteria as bioresources of chemically active and modified compounds and their use in biofuel.

Cyanobacteria are considered the oldest living organisms, which have existed for 3.5 billion years and flourish in every possible niche on Earth, even in extreme habitats, such as high-salinity ponds, hot springs, and polar regions [32–37]. They are among the key players of ecosystems, along with fungi and protozoans, providing crucial services in primary production, decomposition, and nutrient cycling. They are good bioindicators in aquatic environments, especially in coastal and brackish water. They flourish in nutrient-rich wastewater in various morphological features, such as unicellular, filamentous, or colonial forms, forming a mat-like structure [38–42]. Despite being an efficient global sink for atmospheric carbon through the photosynthetic process and natural nitrogen fixation they are also utilized as bio-agents for aquatic pollutant removal [43–46]. Recently, it has been reported that cyanobacterial growth significantly ameliorates the negative effects of herbicides and pesticides [47] in aquatic and terrestrial ecosystems. Besides being rich sources of chemical metabolites, such as carbohydrates, amino acids, and lipids, cyanobacteria do foster other chemical compounds, such as pigments and anti-oxidants, that can cure several pathological conditions [48,49]. Furthermore, cyanobacterial metabolites, especially carbohydrates, can be utilized to produce valuable biochemical derived biopolymers,



biofertilizers, biofuels, nutraceuticals, and enzymes [50,51]. Their extraordinary richness in biomolecular diversity makes them the most promising biofuel agents [52]. Different species of cyanobacteria can be utilized to produce different fuels, such as cellulosic ethanol, biodiesel, biogas, and hydrogen [53]. The high productivity of primary and secondary metabolites in cyanobacteria is due to their potential to assimilate nitrogen and phosphorus in bioactive format from the environment, producing sufficient biomass during a limited growth period, as compared to other land plants or crops [54]. Such a high turnover of biomass is utilized by the neutraceutical and biorefinery industries, and leftover biomass is used as a primary substrate for the production of biofuel. Cyanobacterial farming does not demand land for production and is quickly grown on the aquatic system, including seawater, wastewater, and fresh water. They are the potential source of third-generation feedstock, converting solar energy into green fuels. The appropriate selection of cyanobacterial species matching the local environment is a crucial step for maximizing biomass production for economic and commercial success. The success of suitable cyanobacterial species according to the local environment lies behind their balanced cell stoichiometry and optimization at minimum operational cost. Furthermore, wastewater is the cheapest and best growth medium for algal biomass production because of its high nutritional value [55]. Recent scientific reports also indicate that algae-based wastewater remediation can also produce biofuel at a commercial level. Because of the bioremediation property of algae, they are employed in wastewater treatment from industrial and domestic sources, including slaughterhouses, textile pharmaceuticals, and agro-industries [56,57]. The efficiency of biofuel depends on many physical and chemical properties, such as oxidation constant, cetane number, cold flow, flash point, cloud point, pour point, etc. [58]. While selecting the most suitable strain/species of cyanobacteria for commercial biofuel production, one must consider all these properties along with the environmental condition. Recently, *Fremyella diplosiphon* has been reported to have transesterification of lipids in biodiesel, increasing the cetane number and oxidation constant above the threshold standards of biofuel [59]. In addition, it increases the density, viscosity, plugging point of the iodine cold filter, and cloud and pour points above the lowest acceptable level. Some other cyanobacterial species, such as *Cyanobium* sp., *Limnothrix* sp., and *Nostoc* sp., have been tested and commercialized in biodiesel production. In particular, *Limnothrix* has been shown to provide the optimum lipid profile with an increased abundance of C16:0 [60]. Many filamentous cyanobacteria are reported to produce high-valued chemicals, such as limonene, farnesene, and linalool. After the extraction of high-valued compounds, the residual biomass may undergo biological fermentation or transesterification for biofuel production [61].

## 2. Cyanobacteria as Potential Feedstocks for Biofuel Production

For biofuel synthesis, feedstock selection is the most important factor determining the lipid content and contributes to three-quarters of the total biofuel production. Table 1 shows how biofuel has evolved from first-generation to fourth-generation fuel production under different processes. Feedstocks can be categorized into five different classes based on the origin of the biofuel.

**Table 1.** Evolution of biofuel from conventional to fourth generation biofuel.

| Type | Nature | Merits/Demerits | References |
|---|---|---|---|
| Conventional Energy Sources | Wood and plant residues (solid fuel) | Undergoes incomplete combustion; produces $CO$, $CO_2$, $SO_2$, $NO_2$, and particulate material, which are injurious to health and environment | [62] |
| | First generation—Biofuel derived from edible plants, such as sugarcane | Competes with edible crops, resulting in the high price of eatable items as well as feedstocks | [63] |
| | Second generation—Biofuel derived from non-edible parts of the plants, includes agricultural waste and switch grasses | Demands excessive use of land, water, chemical fertilizers, and pesticides; non-fuel parts discarded, causing a disposal issue | [64] |

**Table 1.** *Cont.*

| Type | Nature | Merits/Demerits | References |
|---|---|---|---|
| Advanced Energy Sources | Third generation—Biofuel derived with the help of traditional microorganisms (algae, yeast, and bacteria) | Does not compete with food crops; no demands of land, fertilizers, and pesticides; minimum use of land and water bodies | [65] |
| | Fourth generation—Biofuel derived with the help of genetically modified microorganisms with targeted efficiency | An extensive increase in biofuel production due to the modification of targeted genes of the microbial cells | [66] |

Cost and benefit analysis indicates that first and second generation feedstocks demand significant agricultural land and other costlier resources that seriously affect food production in the agriculture sector. Furthermore, careful calculation and research in several reports have also pointed out that biodiesel production from third- and fourth-generation feedstocks has higher production costs than petroleum-derived diesel. Hence, only second-generation biodiesel production at the commercial level is currently feasible regarding cost and feedstock sustainability [67–69]. However, more than 95% of biodiesel production worldwide is from first-generation feedstocks, which are highly convenient as well as viable in those regions where resources for agriculture, such as land and water are available in surplus amounts [70–72]. However, alternative strategies such as transesterification (in *Nostoc punctriforme*), fermentation (in *Synechococcus* strain, *Gloeocapsa alpicola*, *Anabaena* sp.) and co-digestion with manure (in *Lyngbya* sp.) are also commercially viable for biofuel production [1,65,66,73–75] (Table 2).

**Table 2.** Biodesigned cyanobacterial strains for biofuel production.

| Cyanobacteria Species/Strain | Product(s) | Biosynthetic Pathway/ Mechanism |
|---|---|---|
| *Spirulina platensis*, *Anacystis nidulans* | Alkanes (C15–C17) | Photosynthesis |
| *Synechocystis* sp. | Butanol | Fermentation |
| *Nostoc punctriforme* | Biodisel | Transesterification |
| *Synechococcus* strain | Bioethanol | Fermentation |
| *Gloeocapsa alpicola*, *Anabaena* sp. | Biohydrogen | Fermentation |
| *Lyngbya* sp. | Biogas | Co-digestion with manure |

On the other hand, biodiesel's preparation from edible oil crops will share the limited available cropland, resulting in a shortage of food supply. However, diesel from non-edible oil crops will not negatively impact food production and supply but will adversely affect the land and water resources. Researchers have already calculated the high cost of biodiesel synthesis from non-edible oils compared to petro-diesel. The natural bioavailability of algae, the most promising feedstock, has the potential to fulfill the demand for renewable energy–based fuel without any aid. Third-generation feedstocks may be the most widespread, as they include waste remnants of cooking oil, animal fats, plant fats, and effluent palm oil factories, which can be used for biodiesel production. Food processing industries do utilize a large amount of vegetable oil. In fact, biodiesel derived from oil-based waste is cheaper and more eco-friendly because no land or water resources are used and there is zero interference in the food chain supply.

## 3. Role of Cyanobacteria in High-Valued Biofuel

Cyanobacteria produce a wide range of metabolic products that are efficient substrates for biofuel production [74–77]. Stored macromolecules in cyanobacterial biomass are carbohydrates, lipid/fatty acids and proteins having the respective caloric value depending on the end product. Carbohydrate has calorific value of 26.72 and 32.5 kJ/g when its

end products are bioethanol and biobutanol respectively. On the other hand, lipid/fatty acids have calorific value of 37.27 kJ/g producing the biodiesel. Besides cultivation, other steps, such as harvesting, extraction, and fuel production, are also money demanding steps. However, cyanobacteria are cultivated as biofilm, which curtails the costlier biomass harvesting step, reducing the total capital input (Table 3).

**Table 3.** Chemical composition of biofuel derived from cyanobacteria [77].

| Nature of Stored Macromolecules | Calorific Value (kJ/g) | Derived Fuel |
| --- | --- | --- |
| Carbohydrates | 26.72 | Bioethanol |
| | 32.5 | Biobutanol |
| Lipid/fatty acids | 37.27 | Biodiesel |
| Carbohydrates and proteins | 150.00 | Biohydrogen |
| | 43.00 | Biogas |

Furthermore, chemical flocculation [76,77] methods using inorganic (i.e., lime and aluminum sulfate) and organic compounds (i.e., chitosan and polyelectrolyte) increase the harvesting of cyanobacterial biomass significantly. Many cyanobacterial species such as *Spirulina platensis*, *Anabaena*, and *Microcystis* have gas vesicles inside their cytoplasm that impart to them a type of natural flotation property, facilitating cheaper harvesting of cyanobacterial biomass. The supplementary addition of NaCl to *Spirulina* biofilm results in the flotation of up to 80% of the total biomass within a few hours, offering a cheaper and more effective harvesting approach. After harvesting, the drying and dewatering of the biomass is the next step for biofuel production [78,79]. The following Figure 1 gives a clear-cut illustration of the steps from microalgal growth optimization to biofuel production, including strain development and the possibility of incorporating the cultivation of algae with an existing setup of wastewater treatment. Inorganic carbon uptake is an essential process for the highest rate of the production of fuels with biological origin from cyanobacteria [80].

### 3.1. Biodiesel from Cyanobacteria

Biochemically, biodiesel comprises long-chain fatty acids of mono-alkyl ester, the best alternative to petroleum-derived diesel. The initial processes of the generation of conventional jet fuel and biodiesel are synthesized by transesterification, where lipids or bio-oils, i.e., triglycerides, are serially converted into esters via diglycerides and monoglycerides. Significant components of biodiesel, such as glycerol and fatty acid methyl esters (FAMEs), can be obtained from the methanol and fatty acids (FAs) in the presence of a strong acid or base catalyst [81–83]. This synthesis pathway results in the end product of lipids, which is significantly beneficial, as lipids store more energy in comparison to carbohydrates [84]. Moreover, esters of biodiesel can be easily formed from cellular lipids. Biofuel can be directly produced from the extracts of lipids by blending the cyanobacterial biomass with alcohol and a heterogeneous catalyst under high temperature. In this process, cells of cyanobacteria undergo reactions with methanol in the presence of strong acid catalyst such as sulfuric acid inside a microwave reactor resulting in the transesterification of FAs followed by chloroform: methanol phase separation [85,86]. When total lipids undergo direct transesterification, the FA profile is enriched in total, as observed in the cyanobacterial biomass of *Synechocystis* sp. and *Synechococcus elongates*. The above procedure of chemical preparation results in high economic and logistical benefits as compared to plant-derived biodiesel derived from terrestrial crops such as soybean and corn. It is also possible to produce biodiesel from algae farmed in ponds on a very large scale when compared with the yield of biofuel from fuel crops, e.g., soya or rapeseed [87–91]. In fact, the above way of deriving cyanobacterial biofuel is advantageous for the environment, as it presents low

sulfur emissions, zero production of aromatic hydrocarbons, the release of oxygen, and good combustion capacity.

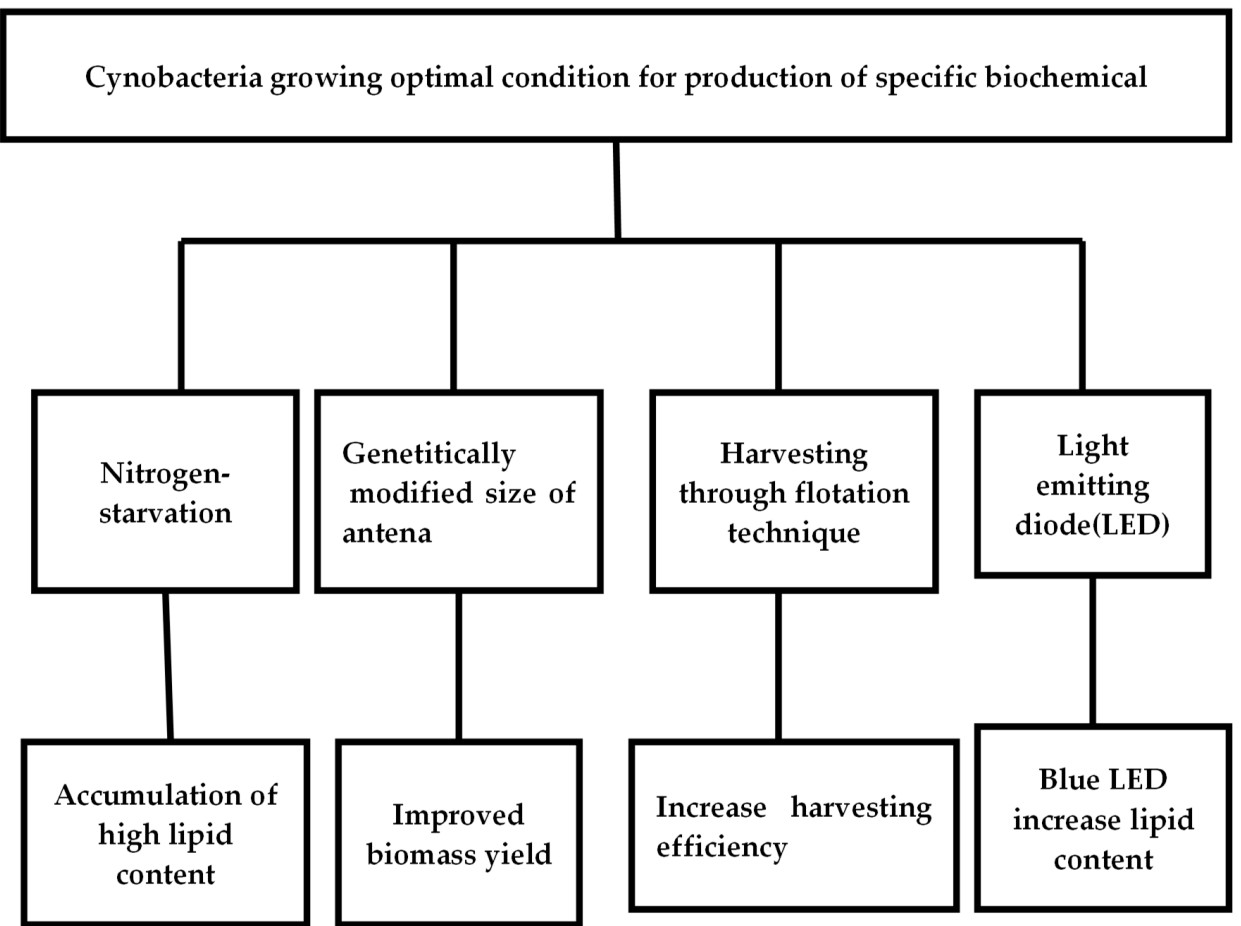

**Figure 1.** Modified cultivation and harvesting technique of cyanobacterial biomass.

Recently, some studies have focused on the development of innovative techniques for the extraction of crucial components of biofuel, such as methyl ester from algae, which offers compatibility with conventional diesel engines. However, the variation in the degree of competitiveness of cyanobacterial triacylglycerides in relation to other lipids derived from algal biomass hinders the commercial establishment of biofuel in the transportation industry [85]. Different species of cyanobacteria contain varying concentrations of fats, reaching a peak of 60% of their overall weight.

Scientific interest in algal oil is not a new trend, but its application for biofuel production is a recent trend in research communities. Algal oil, especially that from macroalgae, such as seaweed, is principally used in cosmetic industries. The various chemical compositions of different algal species indicate an average presence of 40% fat of their overall mass in most of the cyanobacterial species [86]; lipid and FA concentrations of microalgae may change depending on the culture conditions. In algal oil, saturated and monounsaturated FAs, such as oleic (18:1), palmitic (16:0), stearic (18:0), linoleic (18:2), and caproic acids (C6:0), are reported to be found. Algae accumulate 30–80% of lipids, usually in the form of 90–95% triacylglycerides. Wastewater cultivation is considered cost-effective in increasing the production of biomass and altering the concentration of FAs and lipid composition. Among the algal strains, *Chlorella* produces FAs in the range of C16–C18, which are considered suitable for the production of biodiesel, having similar properties as fossil-based biodiesel [87–89].

### 3.2. Bioethanol from Cyanobacteria

Bioethanol is synthesized by the fermentation of carbohydrates extracted from the algae or plants such as corn, sugarcane, wheat, and lignocellulosic biomass. First-generation bioethanol production, which is the traditional way of alcoholic fermentation, utilizes food crops as feedstocks (e.g., wheat, corn, potatoes, beets, sugarcane). These crops are excellent feedstocks for fermentation as they have very high indexes of starch and sugar and are easily available in the agro-sector. However, as the human population increases, putting more and more burdens on the limited agricultural land, serious concern arises over the fuel generation from food crops. Therefore, different sources, especially from the biomass of non-edible crops such as lignocellulosic materials and algae, are being examined as feedstocks for sustainable bioethanol production at the commercial level. Therefore, bioethanol generation can be carried out by utilizing feedstocks from non-food crops. However, specific cyanobacterial strains producing complex carbohydrates also result in the production of synthetic gas and bioethanol, similar to non-edible crops.

Carbohydrates extracted from the biomass of cyanobacteria can be changed to bioethanol by following the processes of hydrolysis and fermentation, which have various steps. Cytosolic sugars, often without oxygen, are channelized in glycolysis to produce free energy through fermentation, generating ethanol and $CO_2$. As a source of fuel, bioethanol is commonly considered, as it has wide applications in existing diesel engines without any significant modification. The hydrolysis of *Synechococcus* sp. biomass through an enzymatic process followed by fermentation with the help of yeast tremendously increases the yield of ethanol quantity. Since glycogen, the storage form of carbohydrates, requires less storage volume inside the cell, it is preferred over the other forms of carbohydrates as a feedstock for bioethanol production [90–92]. Approximately 86% of ethanol production can be obtained through the fermentation of *Synechococcus* sp. with the help of yeast *Saccharomyces cerevisiae*. Cyanobacteria can be subjected to chemical hydrolysis for the lipid extraction process, thus fulfilling the dual purpose of recovering fermented carbohydrates and fats from the biomass. The evaluation of fat content in microalgae *Tribonema* sp. before and after hydrolysis indicates a 25% increase in its production. In addition to this, dark-fermentation and photo-fermentation processes are also employed to generate ethanol, the efficiency of which relies on the metabolic requirements of the cyanobacteria [93,94]. Algal species such as *Chlamydomonas*, *Spirulina*, *Euglena*, *Chlorella*, *Scenedesmus*, and *Dunaliella* have been vastly investigated for the production of bioethanol. The sugar content of algae can also be employed to generate biobutanol, biomethane, biogas, and syngas [95–100]. However, some countries manage to produce bioethanol from the feedstock of food crops. Brazil is totally dependent on sugarcane for bioethanol production to fulfill its requirement to a large extent.

Most of the conducted studies show that bioethanol production efficiency improves when cyanobacteria contain a lesser amount of lignocellulosic material as compared to higher plants. It shows that a lack of lignocellulosic biochemical can enhance the fermentation process [101].

### 3.3. Biobutanol from Cyanobacteria

In past few decades, direct production of short chain fuels like butanol has increased and providing an efficient way to the large scale production of technologies for alternative energy. Butanol is a 4-carbon alcohol ($C_4H_9OH$), a special bulk chemical and excellent blend in fossil fuel. Cyanobacteria such as *Oscillatoria obscura* and *Lyngba limnetica* are being used for biobutanol production at the commercial level by using *Clostridium beijerinckii* ATCC 35,702 as the fermenting microorganism in the presence of glucose. The productivity of biobutanol from these cyanobacteria is found to be 1.565 g/L, obtained with the supplement of glucose in the batch mode condition [89]. Furthermore, the genetically modified cyanobacteria *Synechocystis* PCC6803 sp. have been observed to emit fewer GHGs (3.1 kg $CO_2$ eq/kg biobutanol) into the environment. However, genetically engineered microorganisms are being tested for sustainable biobutanol production [102–106].

### 3.4. Biohydrogen from Cyanobacteria

Biohydrogen is another fuel source that is renewable and yields $H_2O$ as the primary waste from its combustion reaction. It has been revealed that *Anabaena* spp. produces a high quantity of $H_2$ [107,108]. It is known as a clean biofuel, having the highest energy density and eco-friendly production. Cyanobacteria perform biophotolysis of water molecules in the presence of sunlight for biohydrogen production. However, genetically modified cyanobacterial species have a greater potential for biohydrogen production, especially in fuel cells, hydrocarbon liquefaction, and excellent-quality heavy oils [107–110]. Biohydrogen can also be generated by cyanobacteria grown in an $N_2$-deficient environment through the reversible activity of hydrogenase enzymes. Hydrogen production from cyanobacteria requires the presence of nitrogen-fixing strains having bidirectional hydrogenase, hydrogenase, and nitrogenase enzymes. It may be noted that non-heterocystous cyanobacteria are less efficient at $H_2$ gas generation than heterocysts.

Among the different biohydrogen production techniques, biophotolysis, a dark-fermentation method, directly or indirectly uses carbohydrates from cyanobacteria to synthesize biohydrogen [111]. It is noted that a maximum of twelve moles of molecular hydrogen is produced per mole of glucose, as shown in the following formula:

$$C_6H_{12}O_6 + 6H_2O \rightarrow 6CO_2 + 12H_2$$

The microbial bioelectric fuel cell is the greenest and the most sustainable biohydrogen production method for eco-friendly green fuel production. In most microbial fuel cells (MFCs), anodes are constructed from a cyanobacterial strain for hydrogen production, and cathodes are constructed from microalgae for oxygen production. They exploit microorganisms for the production of biohydrogen and are necessary for the function of fuel cells. The most significant benefits of these biological processes for energy production are the bioremediation activities.

### 3.5. Biogas Production from Cyanobacteria Waste

Cyanobacteria also produce gaseous fuels such as syngas for biofuel purposes. The residual biomass of cyanobacteria is converted into biogas through several conversion pathways. These processes pass through hydrolysis and fermentation, converting soluble glucose constituents into alcohols and other intermediate biogas products. The biogas produced in this way contains a mixture of methane, carbon dioxide, and hydrogen [107]. Some trace elements in cyanobacteria, along with nutrients such as proteins, lipids, and carbohydrates, can stimulate the process of methanogenesis for biogas formation. Biogas has been shown to depend on the quality of biomass and the pretreatment process. Among cyanobacterial strains evaluated for biogas production, *Spirulina* species has a conversion efficiency of up to 59% at 35 °C. Wastepaper sludge pretreated with cyanobacterial *Phormidium valderianum* strain enhances biodegradation and improves methane production efficiency. Balancing the carbon and nitrogen ratio increases cellulase activity and produces significant amounts of methane. Cyanobacteria also degrade harmful bioconstituents, such as cyanides, and convert them to methane, a unique biological capability found in certain cyanobacteria, such as *Anabaena* sp. *Arthrospira platensisis* is also reported to remove carbon dioxide from sewage sludge [112,113]. These supplementary techniques are expected to decrease the production cost of biofuel, which is viable for the bioenergy process.

## 4. Present Developments and Future Challenges

The energy budget always remains at the center of each country's economic policy. There are many examples of conflicts due to the allocation of energy and sources of fossil fuels. Success in the production of a new generation of biofuel will make a country self-dependent in the energy sector and may lessen these conflicts. However, there are several challenges in achieving success at the commercial level in biofuel generation, such as high investment costs for new biodiesel industries. Researchers have succeeded in the

selection of feedstock, especially for cyanobacterial biomass and non-crop biomass, but the setup of a biodiesel industry demands costlier construction and huge expenditures on infrastructure, such as transmission, grid interface, integration systems, and the collection system. However, there are still challenges, such as huge expenditures associated with biomass generation from algae and the post-harvesting process, which question the economic viability of biofuel generation.

## 5. Conclusions

The cultivation of algae in wastewater provides remarkable dual alternatives to deal with higher operational expenditures: biofuel production and effective wastewater treatment. It must be noted that algal biomass produces not only lipids and carbohydrates but also a variety of proteins and pigments, which may enhance its value. In this regard, a biorefinery concept will be more viable when a variety of commercial products along with biofuel can be generated from a single process. Moreover, the efficiency of the nutrient removal by microalgae can be enhanced by employing binary cultures that may consist of microalgae–microalgae, microalgae–bacteria, or microalgae–fungi associations. These associations will improve the flocculation ability of biomass, while preventing contamination and improvising microalgae's natural metabolite content. However, detailed investigations are required to study the effect derived from the fusion of wastewater treatment and biomass production and its impact on the post-harvesting process. Keeping an open eye on the advancement in algal research, especially in cyanobacteria, will facilitate the exploration of fifth-generation biofuel, but it needs deep investigations and sufficient funding from the government to achieve commercial robustness in the biofuel industry.

**Author Contributions:** Conceptualization, J.P.R. and K.B.S.; methodology, K.; software, S.V.; validation, R.L. and J.P.R.; formal analysis, K.B.S.; investigation, J.P.R.; resources, K.; data curation, S.V.; writing—original draft preparation, J.P.R.; writing—review and editing, K. visualization, R.L.; supervision, J.P.R.; project administration, J.P.R.; without any funding acquisition. All authors have read and agreed to the published version of the manuscript.

**Funding:** This research received no external funding.

**Institutional Review Board Statement:** Not applicable.

**Informed Consent Statement:** Not applicable.

**Data Availability Statement:** Not applicable.

**Conflicts of Interest:** The authors declare no conflict of interest.

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
