# Peer review of "Current Issues and Developments in Cyanobacteria-Derived Biofuel as a Potential Source of Energy for Sustainable Future"

_sustainability, doi:10.3390/su151310439_

Round 1

Reviewer 1 Report

The authors have reviewed the application of cyanobacteria. They have analysed the potential for using it as a sustainable fuel. The following comments are raised and need justification.

1.      The title of the manuscript needs to be revised for a better understanding of the reader.

2.      Keywords should be revised. Authors should not use abbreviations in the keywords. To improve the article’s visibility during online searching, the authors should not repeat the words in the title again in the keywords.

3.      Revise the abstract. The abstract should be stand-alone with the review purpose, the relevance or importance of the review, and the primary outcomes.

4.      The Introduction should give the nature and scope of the problem studied in the article as clearly as possible. The Introduction written by the author is too unpretentious.

5.      Refer to and cite the following research articles to improve the content of the review. https://doi.org/10.1016/j.eti.2021.101389; DOI: 10.1007/s11356-021-16163-9; https://doi.org/10.1016/j.enconman.2019.06.069; https://doi.org/10.1016/j.enconman.2016.12.054;

6.      The first time authors use an acronym, write the phrase in full and place it in parentheses immediately after it (e.g. GHG, FAME, FAS, etc.). You can then use the acronym throughout the rest of the text, including the abstract, along with the list of abbreviations given in separately.

7.      The title of the Table needs to be revised for a better understanding of the authors.

8.      Revise the statements given in section 4 (from line number 180 to 192).

9.      Refer to and cite the following research articles to improve the content of the biodiesel review: Optimization of various parameters on Botryococcus braunii for biodiesel production using nano cao catalyst; Vishnupriya MV, Ramesh K, Sivakumar P, Vijayakumar B. Biodiesel production from oleaginous microbes - A review; Deepalakshmi S, Sivalingam A, Thirumarimurugan M, Yasvanthrajan N, Sivakumar P. In-situ transesterification of algal biomass for the production of biodiesel;   https://doi.org/10.1080/15567036.2018.1549124; https://doi.org/10.1007/s11356-022-23163-w; https://doi.org/10.1007/s11356-022-23163-w; https://doi.org/10.1016/j.clce.2022.100038; https://doi.org/10.1007/s11356-020-07984-1; https://doi.org/10.1080/00986445.2018.1494582; https://doi.org/10.1016/j.enconman.2014.09.019; https://doi.org/10.1016/j.fuel.2012.01.046;

10.  In section 7 authors have to discuss electrodes and membranes used in hydrogen production along with the parameters affecting the process. (Refer: Sivakumar P, Ilango K, Praveena N, Sircar A, Balasubramanian R, Sakthi Saravanan A, et al. Algal Fuel Cell. In: Jacob-Lopes E, editor. Microalgae Biotechnol. 1st ed., London, UK: IntechOPen Limited; 2018, p. 92–103.)

11.  It is suggested that the author should compare some typical works in this field and give their suggestion on it.

12.  The authors should not use personal pronouns in the manuscript ( e.g. We).

13.  The authors can also include a subsection on economic benefits. It is suggested that the author can compare some typical works in this field.

14.  The authors have to incorporate a subsection on challenges and ways forward above the conclusion.

15.  Make the conclusion section separate with a single paragraph having 150 to 200 words. Rewrite the conclusion with a restatement of the review, a summary of your key arguments and/or findings, and a short discussion of the implications of the review and the future study required.

Several typographical, grammar and language errors are present in the manuscript. The manuscript needs major language corrections.

Author Response

Reviewer 1: Response

The following comments are raised and need justification.

Comment 1- The title of the manuscript needs to be revised for a better understanding of the reader.

Reply: Thanks for your valued comment. The title of the MS has been changed as per your suggestion.

Comment 2- Keywords should be revised. Authors should not use abbreviations in the keywords. To improve the article’s visibility during online searching, the authors should not repeat the words in the title again in the keywords.

Reply: Thanks for your comment. Infact, we were not aware of the fact. We have changed the key words. The set of new key words are not present in the title of the paper and we have removed abbreviations also.

Comment 3- Revise the abstract. The abstract should be stand-alone with the review purpose, the relevance or importance of the review, and the primary outcomes.

Reply: Dear reviewer, as per your suggestion, we have formulated the abstract again in the light of your comment.

Comment 4- The Introduction should give the nature and scope of the problem studied in the article as clearly as possible. The Introduction written by the author is too unpretentious.

Reply: Yes, we are fully agreed with your comment. The introduction part has been rewritten as per your suggestion. Thanks for such a valued comment.

Comment 5- Refer to and cite the following research articles to improve the content of the review. https://doi.org/10.1016/j.eti.2021.101389;

DOI: 10.1007/s11356-021-16163-9;

https://doi.org/10.1016/j.enconman.2019.06.069;

https://doi.org/10.1016/j.enconman.2016.12.054;

Reply: We consulted the above mentioned article and necessary statements have been incorporated with respective references. Thanks for your suggestion.

Comment 6- The first time authors use an acronym, write the phrase in full and place it in parentheses immediately after it (e.g. GHG, FAME, FAS, etc.). You can then use the acronym throughout the rest of the text, including the abstract, along with the list of abbreviations given in separately.

Reply: The appropriate changes as per your suggestions have been followed throughout the whole MS. Thanks for pointing out the mistake.

Comment 7-The title of the Table needs to be revised for a better understanding of the authors.

Reply: Yes, We have modified the title of the Table.

Comment 8- Revise the statements given in section 4 (from line number180 to 192).

Reply: According to your suggestion, the above portion has been revised again.

Comment 9- Refer to and cite the following research articles to improve the content of the biodiesel review: Optimization of various parameters on Botryococcus braunii for biodiesel production using nanocatalyst; Vishnupriya MV, Ramesh K, Sivakumar P,Vijayakumar B. Biodiesel production from oleaginous microbes – A review;

 Deepalakshmi S, Sivalingam A, Thirumarimurugan M,Yasvanthrajan N, Sivakumar P. In-situ transesterification of algal biomass for the production of biodiesel; https://doi.org/10.1080/15567036.2018.1549124;https://doi.org/10.1007/s11356-022-23163-w;https://doi.org/10.1007/s11356-022-23163-w;https://doi.org/10.1016/j.clce.2022.100038;https://doi.org/10.1007/s11356-020-07984-1;https://doi.org/10.1080/00986445.2018.1494582;https://doi.org/10.1016/j.enconman.2014.09.019;https://doi.org/10.1016/j.fuel.2012.01.046; 10. In section 7 authors have to discuss electrodes and membranes used in hydrogen production along with the parameters affecting the process. (Refer: Sivakumar P, Ilango K,Praveena N, Sircar A, Balasubramanian R, Sakthi Saravanan A, etal. Algal Fuel Cell. In: Jacob-Lopes E, editor. MicroalgaeBiotechnol. 1st ed., London, UK: IntechOPen Limited; 2018, p. 92–103.)

Reply: Really we offer our thanks to you for such meaningful comments. We have consulted the above mentioned paper and have incorporated the statements wherever it was necessary. Once again we thank you for suggesting the related papers for our review.

Comment 11. It is suggested that the author should compare some typical works in this field and give their suggestion on it.

Reply: Yes, during the modification and rewriting of the MS, we have incorporated the things in different sections.

Comment 12. The authors should not use personal pronouns in the manuscript ( e.g. We).

Reply: Thanks, really we were not aware of the fact while writing the MS. However, now we have replaced the pronoun “we” form the MS as per your suggestion.

Comment 13. The authors can also include a subsection on economic benefits. It is suggested that the author can compare some typical works in this field.

Reply: We have included as per your suggestion in the MS.

Comment 14. The authors have to incorporate a subsection on challenges and ways forward above the conclusion.

Reply: As per your suggestion, we have created a sub section of “ Present developments and Future challenges”. Thanks for your comment.

Comment 15. Make the conclusion section separate with a single paragraph having 150 to 200 words. Rewrite the conclusion with are statement of the review, a summary of your key argument sand/or findings, and a short discussion of the implications of their view and the future study required.

Reply: As per your comments, we have formulated the conclusion having the statement of the review, summary and implications on future research.

Reviewer 2 Report

Dear authors, the manuscript "Prospects of Cyanobacterial Biofuels as Potential Sources of the "New Generation Sustainable fuels of the Future" is quite interesting and worth investigation. Please see some comments below:

1- Please double-check enghish grammar and formatting (e.g. remove terms as we; Cyanobium sp.

2- Introduction should be more concise 

3- I did like the structure (4, 5...), nevertheless I thenk they are sub-item (3)

4- Personally speaking, you should emphasize the cultivation conditions, fuel sources mainly lipid extractions, reactions, drawbacks and advantages.

Regards

1- Please double-check enghish grammar and formatting (e.g. remove terms as we; Cyanobium sp.

Author Response

Reviewer 2: Response

Please see some comments below:

Comment 1- Please double-check English grammar and formatting (e.g. remove terms as we; Cyanobium sp.

Reply- Dear reviewer, the whole MS has been rewritten taking care of English grammar and formatting as per your suggestion.

Comment 2 - Introduction should be more concise.

Reply: The introduction has been rewritten in concise way as per your suggestion

Comment 3- I did like the structure (4, 5...), nevertheless I think they are sub-item (3)

Reply: It was our silly mistakes. We have changed it in the form of 3.1, 3.2 like in place of 4,5, etc. Thanks for the valuable comments.

Comment 4- Personally speaking, you should emphasize the cultivation conditions; fuel sources mainly lipid extractions, reactions, drawbacks and advantages.

Reply: As per your suggestion, the cultivation conditions; lipid extractions, its reactions, drawbacks and advantages has been mentioned in different sections of MS

Reviewer 3 Report

The proposed text is an interesting collection of documents, although it should be deepened better to be a real review.

First of all I recommend writing the introduction better:

- Some parts (e.g. lines 27-33) sound more like a novel than a scientific article... not the appropriate language

- I suggest we take a look at the various biofuels... on line 49 we move too quickly from fossil fuels to cyanobacteria: It seems that there are no other biofuel production techniques

The following paragraphs often lack citations of important concepts:

e.g. 132-134 lines and 171-173 lines

Tables 2 and 3 are interesting, but both refer to a single article... in a review like this it would be appropriate to look for more sources and more data available.

in line 174 we speak of a figure who, however, does not appear

I do not understand line 193-194 because it is referred to commercial microwave

Why does paragraph 5 talk about sources and not bioethanol from cyanobacteria like the other paragraphs?

Biogas production from cyanobacteria from waste has only one quotation... is there anything else in the literature on the subject?

To give more solidity to the review, it would be appropriate to insert data from literature on the yields of the various production processes of the different biofuels

English is good, small narrative errors in sentence construction

Author Response

First of all I recommend writing the introduction better:

Reply: Dear reviewer, we are agreed with your comments about the poor writing of the introduction section. We realized and have rewritten not only the introduction but also the whole MS. Thank for your valuable comment which improved the review to a large extent.

Comment 1- Some parts (e.g. lines 27-33) sound more like a novel than a scientific article... not the appropriate language.

Reply – Yes, such portions have been written again in a very scientific way

Comment 2- I suggest we take a look at the various biofuels... on line 49 we move too quickly from fossil fuels to cyanobacteria: It seems that here are no other biofuel production techniques.

Reply- Yes, it was a little bit abrupt transition from fossil fuels to cyanobacteria. We have modified the section in appropriate way with gradual transition from fossil fuel to biofuel. Infact, the biofuel is derived from the biomass. Biomass may be cyanobacteria, algae, crop and non crop plants but in all cases, it has to be fermented to produce the biofuel. It has been incorporated in table 1 properly after your suggestion.

Comment 3-The following paragraphs often lack citations of important concepts: e.g. 132-134 lines and 171-173 lines.

Reply- The appropriate references have been incorporated. Thanks for your suggestion.

Comment 4-Tables 2 and 3 are interesting, but both refer to a single article... in a review like this it would be appropriate to look for more sources and more data available.

Reply- We are sorry for the single reference. The multiple original references have been mentioned in table 1 and 2 as per your suggestion.

Comment 5- In line 174 we speak of a figure who, however, does not appear.

Reply- We are sorry. The figure has been inserted in the section “3. Role of cyanobacteria in high valued biofuel”. The diagram gives a simplified idea about the different conditions influencing the production of cyanobacterial biomass.

Comment 6- I do not understand line 193-194 because it is referred to commercial microwave.

Reply: Yes, it is actually “Microwave reactor” rather than “Commercial microwave”. Thanks for pointing out the mistake. The same has been replaced and rectified.

Comment 7- Why does paragraph 5 talk about sources and not bioethanol from cyanobacteria like the other paragraphs?

Reply- The above issue has been corrected and discussed properly in the section. Thanks for your comment.

Comment 8- Biogas production from cyanobacteria from waste has only one quotation... is there anything else in the literature on the subject?

Reply- The appropriate references have been inserted and discussed in the above mentioned section. It was our silly mistakes.

Comment 9- To give more solidity to the review, it would be appropriate to insert data from literature on the yields of the various production processes of the different biofuels.

Reply- Yes, we have now revised the MS and mentioned different production process whenever it is required. Kindly have a look on the table 3. Thanks for your comment.  

Round 2

Reviewer 3 Report

Good work